# Biofilm Production Ability, Virulence and Antimicrobial Resistance Genes in *Staphylococcus aureus* from Various Veterinary Hospitals

**DOI:** 10.3390/pathogens9040264

**Published:** 2020-04-04

**Authors:** Lin Chen, Zi-Yun Tang, Shi-Yun Cui, Zhen-Bao Ma, Hua Deng, Wei-Li Kong, Li-Wen Yang, Chao Lin, Wen-Guang Xiong, Zhen-Ling Zeng

**Affiliations:** 1Guangdong Provincial Key Laboratory of Veterinary Drugs Development and Safety Evaluation, National Risk Assessment Laboratory for Antimicrobial Resistance of Animal Original Bacteria, South China Agricultural University, Guangzhou 510642, China; chenl@stu.scau.edu.cn (L.C.); 20182027026@stu.scau.edu.cn (Z.-Y.T.); cuisy@stu.scau.edu.cn (S.-Y.C.); mazhenbao@stu.scau.edu.cn (Z.-B.M.); kongwl@stu.scau.edu.cn (W.-L.K.); yanglw@stu.scau.edu.cn (L.-W.Y.); natsume@stu.scau.edu.cn (C.L.); 2Guangdong Laboratory for Lingnan Modern Agriculture, South China Agricultural University, Guangzhou 510642, China; 3School of Life Science and Engineering, Foshan University, Foshan 528231, China; denghua@fosu.edu.cn

**Keywords:** *Staphylococcus aureus*, prevalence, biofilm, virulence, antimicrobial resistance, veterinary hospital

## Abstract

*Staphylococcus aureus* (*S. aureus*) is one of the most clinically important zoonotic pathogens, but an understanding of the prevalence, biofilm formulation ability, virulence, and antimicrobial resistance genes of *S. aureus* from veterinary hospitals is lacking. By characterizing *S. aureus* in different origins of veterinary hospitals in Guangzhou, China, in 2019, we identified with the presence of *S. aureus* in pets (17.1%), veterinarians (31.7%), airborne dust (19.1%), environmental surfaces (4.3%), and medical device surfaces (10.8%). Multilocus sequence typing (MLST) and *Staphylococcus* protein A (*spa*) typing analyses demonstrated methicillin-sensitive *S. aureus* (MSSA) ST398-t571, MSSA ST188-t189, and methicillin-resistant *S. aureus* (MRSA) ST59-t437 were the most prevalent lineage. *S. aureus* with similar pulsed-field gel electrophoresis (PFGE) types distributed widely in different kinds of samples. The crystal violet straining assays revealed 100% (3/3) of MRSA ST59 and 81.8% (9/11) of MSSA ST188 showed strong biofilm formulation ability, whereas other STs (ST1, ST5, ST7, ST15, ST88, ST398, ST3154 and ST5353) showed weak biofilm production ability. Polymerase chain reaction (PCR) confirmed the most prevalent leucocidin, staphylococcal enterotoxins, *ica* operon, and adhesion genes were *lukD*-*lukE* (49.0%), *sec*-*sel* (15.7%), *icaA*-*icaB*-*icaC*-*icaR* (100.0%), and *fnbB*-*cidA*-*fib*-*ebps*-*eno* (100.0%), respectively. Our study showed that the isolates with strong biofilm production ability had a higher prevalence in *clfA*, *clfB*, *fnbA* and *sdrC* genes compared to the isolates with weak biofilm production ability. Furthermore, 2 ST1-MRSA isolates with *tst* gene and 1 ST88-MSSA isolate with *lukS/F-PV* gene were detected. In conclusion, the clonal dissemination of *S. aureus* of different origins in veterinary hospitals may have occurred; the biofilm production capacity of *S. aureus* is strongly correlated with ST types; some adhesion genes such as *clfA*, *clfB*, *fnbA, and sdrC* may pose an influence on biofilm production ability and the emergence of *lukS/F-PV* and *tst* genes in *S. aureus* from veterinary hospitals should raise our vigilance.

## 1. Introduction

*Staphylococcus aureus* (*S. aureus*) is one of the most clinically important zoonotic pathogens, causing skin and soft tissue infections, bloodstream infections, and even life-threatening diseases with mortality rates higher than those for acquired immunodeficiency syndrome (AIDS), tuberculosis, and viral hepatitis combined [1,2,3]. The human nasal vestibule is the main reservoir of *S*. *aureus*, persistently and intermittently colonizing about 30% and 60% of the human noses, respectively [4]. The colonization is also known to be a risk factor for staphylococcal infections [3]. In addition, *S*. *aureus* strains have been found to survive for long periods on the human skin and the primary mode of transmission of *S*. *aureus* is by direct contact, usually skin-to-skin contact with a colonized or infected individual [1,3]. Studies have indicated that the transmission of *S. aureus* can occur from human to animal and vice versa [5,6]. Alarmingly, numerous cases have been reported that humans may be infected with pet-associated *S. aureus* [7,8], and pets such as dogs and cats can serve as the reservoir of *S. aureus* [9,10]. Moreover, there have been increasing reports of *S*. *aureus* in companion animals, environmental surfaces, and humans from veterinary hospitals or households in the past few years [7,8,9,10,11,12]. 

*S. aureus* has been considered as the most common causative pathogen in surgical site infections in recent years [13]. The ability of *S*. *aureus* to infect humans and animals poses a huge burden on the healthcare system [14]. Molecular typing methods such as pulsed-field gel electrophoresis (PFGE), multilocus sequence typing (MLST), and *Staphylococcus* protein A (*spa*) typing have been widely used to track the sources and transmissions of pathogenic bacteria like *S*. *aureus*, thereby helping to establish the national and global epidemiological data [3]. 

Since Alexander Fleming discovered that penicillin has an inhibitory effect on pathogenic bacteria including *S. aureus*, we have ushered in the ‘antibiotic era’. From 2000 to 2015, global consumption of antibiotics increased significantly, with developing countries such as South Africa, Brazil, India, and China being the most dominate consumers [15,16]. In particular, the major concern is that the consumption of oxazolidinones as the last-resort compounds against methicillin-resistant *Staphylococcus aureus* (MRSA) is rapidly increasing [15]. If there will be no corresponding policy formulation, the global consumption of antibiotics in 2030 would be up to 200% higher than in 2015 [15]. However, a major cause for alarm is the drastic rise of antibiotic resistance of *S. aureus* with the increase in the use of antibiotics [14]. The World Health Organization (WHO) has identified antibiotic resistance as one of the biggest challenges to global health and classified *S. aureus* as a high priority superbug for which research and development of novel antibiotics are needed urgently [17,18]. We should pay attention to the phenomenon that MRSA was constantly detected in companion animals and these strains generally show multidrug-resistance, significantly limiting the treatment options [9,10,11,12,19]. This multidrug resistance could be the result of careless antibiotics usage leading to wide resistance spectrum of MRSA isolated from companion animals [20].

*S. aureus* biofilm formation provides a unique opportunity for persistent infection, antibiotic resistance, and immune evasion [21]. The biofilm-producing pathogen *S*. *aureus* is notorious for its ability to resist treatment by forming biofilms on indwelling medical devices, resulting in persistent infections [22]. Besides, *S. aureus* strains within the biofilm usually show poor response to antibiotics and immune system due to an insufficient drug penetration into the biofilm’s matrix and hindering the recognizable antigens present in bacterial cells, respectively [23]. The main component of *S. aureus* biofilm matrix is the poly-N-acetyl β-1, 6 glucosamine surface polysaccharide (PNAG), synthesized by proteins encoded by the intercellular adhesion *ica* operon [24]. Furthermore, microbial surface components recognizing adhesive matrix molecules (MSCRAMMs) have been shown to function as extracellular matrix components during early biofilm formation [25]. Performing a phenotypic and genotypic characterization of *S. aureus* for biofilm production and analyzing their relationship with antibiotic resistance will be important for understanding the treatment failure of infections caused by *S. aureus*. The crystal violet dye is widely used for in-vitro biofilm quantification due to its low cost and good reproducibility [26,27].

*S. aureus* isolates can increase the pathogenicity by secreting many important virulence factors described as toxins interfering directly with the host cells, with Panton-Valentine Leucocidin (PVL) being one of the most essential, potent, and prevalent [14]. The biofilm described as toxins, toxic shock syndrome toxin (TSST), haemolysin, and staphylococcal enterotoxins (SEs) also played an important role in the life-threatening illness caused by *S. aureus* [21,28]. It is important to note that only a single virulence factor is difficult to function and thus cytotoxicity is often conferred only when more than one factor combined [29]. Therefore, understanding the combination of these virulence genes in *S. aureus* isolates from veterinary hospitals will be vital.

The pet industry is still in its early stage, but has developed rapidly in China. In recent years, the number of dogs and cats has been increasing in developing countries [9]. Keeping pets is becoming a popular lifestyle all over the world. Direct physical contact with pets occurs every day, and therefore poses a potential risk of transmitting the pathogens like *S. aureus* between humans and pets [9]. To the best of our knowledge, there are currently very limited studies aimed to analyze the distribution of *S. aureus* in veterinary hospitals in China. Therefore, we studied the distribution of *S. aureus* in pets, humans, medical device surfaces, environmental surfaces, and airborne dust, and analyzed the risk of transmitting *S. aureus* between humans and pets from veterinary hospitals in Guangzhou, Guangdong, China. Another objective of the study was to determine biofilm production ability as well as the antimicrobial resistance and virulence genes in the *S. aureus* isolates. 

## 2. Results

### 2.1. Prevalence of S. aureus

A total of 51 (13.9%) out of 366 samples were confirmed to be *S. aureus* in the present study. The number of samples collected per veterinary hospital and the prevalence of *S. aureus* in different locations are shown in Table 1. No *S. aureus* isolate was detected in nasal swabs of female veterinarians, infusion pump of medical device surfaces, door handles, switch and drawer of environmental surfaces, and halls with airborne dust. *S. aureus* carriage rate in nasal swabs of male veterinarians (63.6%) was highest and significantly different than the female veterinarians (0%, *p* < 0.05). *S. aureus* carriage rate in skin swabs and nasal swabs of veterinarians was not significantly different (*p* = 0.3). Similarly, there is no significant difference in the detection rates of *S. aureus* in eye swabs and nasal swabs of pets (*p* = 0.8). The carriage rate of *S. aureus* in airborne dust from different locations was not significantly different (*p* = 0.1) and the highest detection rate of *S. aureus* in airborne dust from clinic room (26.7%) was confirmed (Table 1). In addition, surgical forceps, surgical scissors, and weighing scale of medical device surfaces and desktop, chair, cage interior, and ground of environmental surfaces were found with the presence of *S. aureus* (Table 1). The detection rate of *S. aureus* in veterinary hospital 1, veterinary hospital 2, and veterinary hospital 3 was significantly different (*p* < 0.001) (Table 1). The MRSA strains were not isolated from medical device surfaces, environmental surfaces and airborne dust, but were detected in 4 veterinarians and 1 pet (Figure 1).

### 2.2. Antimicrobial Susceptibility and Antibiotic Resistance Genes

The antimicrobial resistance genes and antibiotic resistance profiles of *S. aureus* isolates are shown in Table 2, and Figure 1 and Figure 2. No *S. aureus* isolate was resistant to oxacillin (OXA), doxycycline (DOX), tigecycline (TIG), rifampicin (RIF), vancomycin (VAN), tiamulin (TIA), fosfomycin (FOS), linezolid (LZD) or tedizolid (TZD), but 5 (9.8%), 33 (64.7%), 5 (9.8%), 1 (2.0%), 3 (5.9%), 21 (41.2%), 5 (9.8%), 3 (5.9%), and 3 (5.9%) *S. aureus* isolates demonstrated resistance to amoxicillin (AMO), penicillin (PEN), cefoxitin (FOX), gentamycin (GEN), florfenicol (FFC), erythromycin (ERY), clindamycin (CLI), ciprofloxacin (CIP), and trimethoprim-sulfamethoxazole (SXT), respectively. In addition, 17 (33.3%) *S. aureus* isolates were sensitive to all tested antibiotics (Appendix A). *S. aureus* isolates from veterinarians were resistant to more antimicrobial classes than the isolates from pets, medical device surfaces, environmental surfaces and airborne dust (*p* < 0.001) (Appendix A). The five FOX-resistant *S. aureus* isolates were found to be *mecA*-positive and were thus classified as MRSA isolates. In addition, all MRSA isolates were resistant to AMO and PEN, and thus were confirmed to be MDR isolates (Appendix A). Attention should be paid to the phenomenon that the MRSA isolates were not resistant to OXA. Statistical analysis showed that the MRSA isolates had significantly higher resistance rates to AMO than the methicillin-susceptible *S. aureus* (MSSA) isolates (100.0% vs. 0%, *p* < 0.001), FOX (100.0% vs. 0%, *p* < 0.001), and CLI (60.0% vs. 4.3%, *p* < 0.05) (Appendix A). 

All *S. aureus* isolates were not found with the presence of antibiotic resistance genes of *mecC*, *cfr*, *optrA*, *vanA*, *vanB*, *ermA,* and *aac(6’)-aph(2’’)*. The most prevalent resistance gene was *ermC* (25.5%, 13/51), followed by *ermB* (11.8%, 6/51). The five *S. aureus* isolates carrying *tetK* gene were not resistant to DOX. Four and one *mecA*-positive *S. aureus* isolates were isolated from veterinarians and pets, respectively (Figure 1 and Figure 2).

### 2.3. Virulence Gene Profiles

The frequencies of virulence genes identified in the 51 *S. aureus* isolates are listed in Figure 1. For leukocidin genes, 25 isolates (49.0%, 25/51) harbored *lukD*-*lukE* genes; one MSSA isolate (2.0%, 1/51) from veterinarians harbored *lukS/F-PV* gene. For toxic shock syndrome toxin-1 gene, two MRSA isolates (3.9%, 2/51) from veterinarians and pets had *tst* gene. For SEs genes, 38 isolates (74.5%, 38/51) did not have any tested SEs genes; 8 isolates (15.7%, 8/51) were positive for *sec*-*sel* genes; 7 (13.7%, 7/51) isolates were positive for *seg*-*sei*-*sem*-*sen*-*seo* genes; 6 isolates (11.8%, 6/51) were positive for *sek* gene; 4 isolates (7.8%, 4/51) were positive for *seb* gene; 3 isolates (5.9%, 3/51) were positive for *sea* gene; 3 isolates (5.9%, 3/51) were positive for *seh* gene and one isolate (2.0%, 1/51) were positive for *sed*-*sej* genes. For *ica* operon genes, 51 isolates (100.0%, 51/51) harbored *icA*-*icaB*-*icaC*-*icaR* genes; 33 isolates (64.7%, 33/51) had *icaD* genes. For adhesion genes, 51 isolates (100.0%, 51/51) harbored *fnbB*-*cidA*-*fib*-*ebps*-*eno* genes; 44 isolates (86.3%, 44/51) harbored *fnbA* gene; 43 isolates (84.3%, 43/51) harbored *clfA* gene; 42 isolates (82.4%, 42/51) harbored *cna* gene; 42 isolates (82.4%, 42/51) harbored *sdrE* gene; 40 isolates (78.4%, 40/51) harbored *sdrC* gene; 24 isolates (47.1%, 24/51) harbored *clfB* gene and one isolate (2.0%, 1/51) harbored *sdrD* gene. The virulence genes of *hla*, *hlb*, *hld*, *eta*, *etb*, *see,* and *bap* were not found in all *S. aureus* isolates (Figure 1). Based on the prevalence of the leukocidin genes, the isolates with *lukD*-*lukE* genes had a higher prevalence in pets compared to the other sources. Based on the prevalence of the SEs genes, the *sea* and *seb* genes were detected only in *S. aureus* isolates from veterinarians; the *sec*-*sel* genes were detected in *S. aureus* isolates from all sources; the *sed*-*sej* genes were detected only in *S. aureus* isolates from airborne dust; the *seg*-*sei*-*sem*-*sen*-*seo* genes were detected in *S. aureus* isolates from medical device surfaces, environmental surfaces and airborne dust; the *seh* and *sek* genes were detected in *S. aureus* isolates from veterinarians and pets. Based on the prevalence of the *ica* operon and adhesion genes, the other genes except *bap* and *sdrD* distributed in *S. aureus* isolates from all sources (Figure 1). Except for *ica* operon and adhesion genes, none of ST398 isolates harbored any texted virulence genes (Table 2 and Figure 1). All ST3154 and ST1 *S. aureus* isolates carried *seg*-*sei*-*sem*-*sen*-*seo* and *seh* genes, respectively (Figure 1 and Figure 2). 

### 2.4. Molecular Typing and Biofilm Production Ability of S. aureus

All *S. aureus* isolates represented 10 STs and ST398 (33.3%, 17/51) was the most prevalent, followed by ST188 (21.6%, 11/51), ST3154 (11.8%, 6/51), and ST5353 (9.8%, 5/51) (Table 2 and Figure 2). By *spa* typing, 12 *spa* types were found and the most prevalent *spa* types were t189 and t571 (21.6%, 11/51), followed by t116 (11.8%, 6/51), t7594 (9.8%, 5/51), and t034 (9.8%, 5/51) (Table 2 and Figure 2). One ST1 isolate of veterinarian was confirmed non-typeable by *spa* typing. A strong association was observed between certain STs and *spa* types: ST188 was associated with t189 (100.0%, 11/11); ST3154 was associated with t116 (100.0%, 6/6); ST5353 was associated with t7594 (100.0%, 5/5); ST7 was associated with t803 (100%, 3/3); ST59 was associated with t437 (100.0%, 3/3) and ST398 was primarily associated with t571 (64.7%, 11/17) (Table 2 and Figure 2). All non-ST398 type *S. aureus* isolates were successfully typed by PFGE and 21 major *Sma*I patterns were observed. A strong association was also observed between certain STs and PFGE types: ST188 was primarily associated with Ⅷ (45.5%, 5/11); ST3154 was associated mainly with Ⅰ (50.0%, 3/6) and Ⅱ (50.0%, 3/6); ST5353 was associated mainly with Ⅲ (80.0%, 4/5) (Figure 2). The ST188-t189-Ⅷ pattern was found in 1 isolate from veterinarians and 4 isolates from pets. Attention should be paid to the phenomenon that MRSA isolates (GDX9Z158A and GDX9Z159A) from veterinarians in veterinary hospital 3 shared the similar PFGE types to MRSA isolate (GDY9Z40A) from veterinarians in veterinary hospital 2. Two MRSA isolates (GDY9P40A and GDY9P41A) with similar PFGE types derived from veterinarians and pets in veterinary hospital 2 were confirmed (Figure 2). In addition, *S. aureus* isolates (GDH9P10A from veterinarians, GDH9P1A from pets, GDX9Z11A from environmental surfaces and GDB9Z22A-2 from airborne dust) with similar PFGE types in veterinary hospital 1 and veterinary hospital 3 were also found (Figure 2). In the present study, 100% (3/3) of human-adapted MRSA ST59 and 81.8% (9/11) of MSSA ST188 derived from veterinarians, pets, and airborne dust showed strong biofilm formulation ability, whereas two ST188 isolates (GDX9Z11A and GDX9Z102A) from environmental surfaces and airborne dust in veterinary hospital 3 showed weak biofilm formulation ability (Figure 3). The biofilm production ability of ST188 and ST59 was not significantly different (*p* = 1). Except for ST188 and ST59, all the *S. aureus* isolates with other STs showed weak biofilm production ability (Figure 3). The MRSA ST59 with strong biofilm production ability was resistant to more classes of antibiotics than MRSA ST1 with weak biofilm production ability (*p* < 0.001). However, MSSA ST398 with weak biofilm production ability showed more severe antibiotic resistance than MSSA ST188 with strong biofilm production ability (*p* < 0.001) (Table 2, Figure 2 and Figure 3). The prevalence of adhesion genes among isolates with strong biofilm production ability was as follows: *clfA* (+): 100.0%; *clfB* (+): 66.7%; *fnbA* (+): 100.0%; *fnbB* (+): 100.0%; *cidA* (+): 100.0%; *fib* (+): 100.0%; *cna* (+): 100.0%; *ebps* (+): 100.0%; *eno* (+): 100.0%; *sdrC* (+): 100.0%, and *sdrE* (+): 75.0%. In contrast, the prevalence of *clfA*, *clfB*, *fnbA*, *fnbB*, *cidA*, *fib*, *cna*, *ebps*, *eno*, *sdrC*, *sdrD,* and *sdrE* in isolates with weak biofilm production ability was 79.5%, 41.0%, 82.1%, 100.0%, 100.0%, 100.0%, 84.6%, 100.0%, 100.0%, 71.8%, 2.6%, and 84.6%, respectively. As shown in Figure 1 and Figure 3, the isolates with strong biofilm production ability had a higher prevalence in *clfA*, *clfB*, *fnbA,* and *sdrC* genes compared to the isolates with weak biofilm production ability, but this difference was not statically significant (*p* = 0.65).

## 3. Discussion

*S. aureus* is an important pathogen that can cause diverse serious infections in humans and animals, posing a huge burden on the healthcare system. A relatively high recovery rate (13.9%) of the *S. aureus* isolates was detected in the samples from veterinary hospitals in the present study, with different rates depending on the sample types. In addition, veterinarians and pets were found carrying MRSA. The higher detection rate of *S. aureus* in veterinarians (31.7%) than that of *S. aureus* in other sources supports the notion that *S. aureus* is generally more host-adapted to humans [11]. It is worth noting that *S. aureus* carriage rate in nasal swabs of male veterinarians reached 63.6% and was significantly different to the female veterinarians (0%, *p* < 0.05). It was reported that sex and gender play an active role in the incidence of *S. aureus* bacteremia [30]. The detection rate of *S. aureus* in pets (17.1%) was higher than in previous studies from China [9,31], Zambia [32], Canada [33], and Australia [34], but lower than in the previous study from Greece [35]. The frequency of MRSA in veterinarians (9.8%) was higher in our study than in the previous study from China [9], but lower than in previous studies from Australia [11]. Regarding environmental surfaces, the studies performed in veterinary hospitals [11] and households [10] showed rates of 0% and 46% in MRSA isolates, respectively. The airborne dust and unsterilized medical device also constituted the risk of spreading *S. aureus* [36,37]. The present study indicated the presence of *S. aureus* in medical device surfaces, environmental surfaces, and airborne dust from veterinary hospitals, and *S. aureus* in airborne dust was the most prevalent (19.1%). To our knowledge, the study to isolate and assess *S. aureus* strains derived from medical device surfaces, environmental surfaces, and airborne dust from veterinary hospitals is scarce. Our results added to evidence that the rate of *S. aureus* carriage in the above samples from veterinary hospitals. The contamination level of *S. aureus* in different veterinary hospitals was not the same. This may be owing to different sampling time, different numbers of sample, different geographic location of veterinary hospitals and other factors.

Our results showed that except for PEN (64.7%) and ERY (41.2%), the resistance rates of *S. aureus* to other antibiotics were less than 10%. Similar results were found in *S. aureus* isolated from pets in Tunisia [38]. It is well known that information describing antimicrobial susceptibility patterns of MSSA isolates from veterinary hospitals is limited due to minor significance of MSSA compared with MRSA. Fortunately, all *S. aureus* isolates, including MRSA in this study were susceptible to TIG, LZD, TZD, and VAN, all of which are ‘last resort’ antibiotics for *S. aureus* infections. In the present study, the drug resistance of human-derived *S. aureus* was more serious than that of other sources (*p* < 0.001). This may be due to the higher proportion of MRSA in human samples than the other. The high prevalence of *erm* gene (*ermB* and *ermC*) in ERY-resistant *S. aureus* strains in this study was similar to that in the previous study [9,39]. The antibiotic resistance genes of *cfr*, *optrA*, *vanA*, *vanB,* and *aac(6’)-aph(2’’)* were not detected, which was consistent with their resistant phenotypes. However, attention should be paid to the phenomenon that the *mecA*-carrying *S. aureus* and the *tetK*-carrying *S. aureus* were sensitive to OXA and DOX, respectively. The phenomenon that the strains carried resistance genes but did not show the corresponding antimicrobial resistance was easily ignored, leading to the widespread spread of pathogens and resistance genes [40,41]. 

The results of MLST and *spa* typing showed that ST188-t189 and ST398-t571 were the most prevalent in MSSA isolates, while ST59-t437 was the most prevalent in MRSA isolates. The prevalence of these strains was reported in China in previous studies [39,42,43]. Attention should be paid to the results that ST188-MSSA with similar PFGE types, ST1-MRSA with similar PFGE types, and ST398-MSSA were isolated in the samples of different sources. This indicated that the transmission of these strains among different origins may have occurred and that the pets, environmental surfaces, medical device surfaces, and airborne dust have posed a risk of spreading theses strains to humans. Three ST59-MRSA isolates of veterinarians from two veterinary hospitals with similar PFGE types should also be noted. It was reported that ST59-MRSA isolates can also be discovered from multiple host species [42]. This suggested that ST59-MRSA isolates in this study can infect humans across veterinary hospitals and were at risk of infecting pets. The prevalence of ST188-MSSA, ST398-MSSA, ST1-MRSA, and ST59-MRSA isolates from veterinary hospitals in the present study highlights that the distribution of these strains in China is much wider than expected. ST188 and ST398 might evolve from livestock and ST59 was the predominant clones of community-associated MRSA isolates in China [39,42]. Therefore, we suspected ST188, ST398, and ST59 may have spread from other places such as livestock and community to veterinary hospitals.

In this study, some virulence genes such as *hla*, *hlb*, *hld*, *eta*, *etb*, *see,* and *bap* were not found in all *S. aureus* isolates. However, it should be noted that two ST1-MRSA isolates carrying *tst* gene were isolated from pets and veterinarians, and one ST88-MSSA isolate carrying *lukS/F-PV* gene was confirmed from veterinarians. Up to now, the *tst* gene has not been described in ST1-MRSA isolates from veterinary hospitals in China. The presence of *tst* gene in ST1-MRSA isolates from pets and veterinarians suggested that there was a risk of spreading the virulence gene among MRSA isolates from humans and pets. The ST88 with *lukS/F-PV* gene were reported to cause wound infection and abscesses in humans [44]. The presence of ST88 isolates carrying *lukS/F-PV* gene in veterinarians reminded us that humans could be reservoirs of PVL-positive *S. aureus* isolates and were at risk of infecting with PVL-positive *S. aureus*. Most of the previous studies focused primarily on SEs causing Staphylococcal food poisoning (SFP) [45,46,47,48,49,50]. Our results showed that SEs genes (*sea*, *seb*, *sed*-*sej*, *seg*-*sei*-*sem*-*sen*-*seo*, *seh* and *sek*) expressed specific host specificity. For examples, the *sea* and *seb* genes were detected only in *S. aureus* isolates from veterinarians; the *sed*-*sej* genes were detected only in *S. aureus* isolates from airborne dust; the *seg*-*sei*-*sem*-*sen*-*seo* genes were detected only in *S. aureus* isolates from environmental samples (medical device surface, environmental surface, and airborne dust samples can be classed as environmental samples), and the *seh* and *sek* genes were detected only in *S. aureus* isolates from biological samples (veterinarian and pet samples can be classed as biological samples). Furthermore, SEs genes (*sed*-*sej* and *seg*-*sei*-*sem*-*sen*-*seo*) and SEs genes (*seh*) were significantly associated with ST3154-t116 MSSA and ST1 *S. aureus* isolates, respectively. The *seg*-*sei*-*sem*-*sen*-*seo* genes have been reported in *S. aureus* isolates of several specific ST types but not ST3154 [12,45]. This suggested that the *seg*-*sei*-*sem*-*sen*-*seo* genes are at risk of being further extended to *S. aureus* isolates with other ST types. Attention should be paid to the results that *sec*-*sel* genes were detected in *S. aureus* isolates from all sources in this study. It reminded us that the transmission of *sec*-*sel* genes in *S. aureus* isolates may not affected by host specificity. 

The ability of *S. aureus* to produce biofilms is believed to contribute to food poisoning, antimicrobial resistance and many other problems [21,25]. An important group of virulence factors that initiate early biofilm formation steps are the MSCRAMM proteins, which are encoded by different genes such as adhesion genes [25]. For the texted 13 adhesin genes, our results showed that the detection rate of *clfA***,**
*fnbA*, *fnbB*, *cidA*, *fib*,*can, ebps*, *eno*, *sdrC,* and *sdrE* was more than 78.4%. But the *bap* gene was not detected in all *S. aureus* isolates from veterinary hospitals. Many previous studies reported that the *bap* gene was not detected in any *S. aureus* isolates [25,51,52,53,54]. This suggested that the *bap* gene is rarely prevalent in *S. aureus* according to reported articles. No difference was observed in the prevalence rate of the *fnbB*, *cidA*, *fib*, *ebps,* and *eno* genes among strong biofilm production ability and weak biofilm production ability strains. However, a previous study reported that a highly significant difference was present in the prevalence of *fnbB*, *fib*, *ebps,* and *eno* among these two groups [55]. Our study verified that the isolates with strong biofilm production ability had a higher prevalence in *clfA*, *clfB*, *fnbA,* and *sdrC* genes compared to the isolates with weak biofilm production ability. This was partly consistent with the previous study, indicating that the percentage of *clfA* and *fnbA* in strong biofilm production ability strains to be higher than that in weak biofilm production ability strains [56]. A second group of virulence factors that contribute to biofilm formation is *ica* operon [25]. Our study showed that *icaA*, *icaB*, *icaC,* and *icaR* genes were detected in all *S. aureus* isolates, but the prevalence of *icaD* gene was different between strong biofilm production ability and weak biofilm production ability strains. This differed from an earlier report that the *icaD* gene was detected in all *S. aureus* isolates [25]. Furthermore, this study firstly provided important information on the biofilm formation ability of *S. aureus* isolated from veterinary hospitals in China. A strong association was found between certain STs and biofilm formation ability. All human-adapted MRSA ST59 and 81.8% (9/11) of MSSA ST188 showed strong biofilm formulation ability, whereas other STs with weak biofilm production ability were confirmed. This result was similar to the previous studies in China [42,43]. It was reported that *S. aureus* within the biofilm were poorly responsive to antibiotics and this greatly limited the choice of antibiotics for clinical treatment of *S. aureus* infections [23]. However, our results revealed that *S. aureus* with strong biofilm production ability cannot always show more severe resistance.

## 4. Materials and Methods

### 4.1. Ethical Approval

The study was approved by the South China Agriculture University (SCAU) Animal Ethics Committee. The field sampling protocols, samples collected from veterinary hospitals, strain isolation, and the research were conducted in in strict accordance with Section 20 of the Animal Diseases Act of 1984 (Act No 35 of 1984), Technical Guidelines for Isolation and Identification of Animal Origin *Staphylococcus aureus* (DB51/T 2363-2017) and the Declaration of Helsinki, and were approved with the SCAU Institutional Animal Care and Use Committee and the three veterinary hospitals (Bai Si, Yan Guo Ping and SCAU) in Guangzhou, Guangdong, China.

### 4.2. Sample Collection

A total of 366 samples were randomly collected from 3 veterinary hospitals in Guangzhou, Guangdong, China, from April 2019 to September 2019. The sampling period covered spring, summer, and autumn seasons. The samples included veterinarians (skin swab n = 24; nasal swab n = 17), pets (eye swab n = 41; nasal swab n = 41), medical device surfaces (infusion pump n = 20; weighing scale n = 6; surgical forceps n = 6; surgical scissors n = 5), environmental surfaces (desktop n = 40; chair n = 29; cage interior n = 23; ground n = 17; door handle n = 18; switch n = 7; drawer n = 4), and airborne dust (clinic room n = 30; inpatient department n = 13; hall n = 10; B ultrasound room n = 7; reception room n = 5; corridor n = 3) (Table 1). In short, sterile cotton swabs were used to collect samples from veterinarians, pets, medical device surfaces, and environmental surfaces. The Chromogenic *S. aureus* Agar (Guangdong Huankai Microbial Sci, Guangzhou, China) plates were used to collect airborne dust in different rooms of veterinary hospitals the whole day. These swabs were collected into sterile plastic tubes containing 1.5 mL of 0.85% physiological saline. Then the tubes and the plates were stored on ice during transport to the laboratory at College of Veterinary Medicine of SCAU for further analysis.

### 4.3. Sample Processing and Bacterial Isolation

The 7.5% Sodium Chloride Broth (Guangdong Huankai Microbial Sci, Guangzhou, China) was added to the tubes and incubated for 24 h at 37 °C and 180 rpm/min on a shaker. Then the treated broth was streaked onto Chromogenic *S. aureus* Agar plates. All the Chromogenic *S. aureus* Agar plates were incubated at 37 °C for 36–48 h. Presumptive *S. aureus* pure colonies (characterized by a pink color) were selected for further study. 

### 4.4. DNA Extraction and Molecular Identification of S. aureus

DNA from all presumptive *S. aureus* isolates was extracted using the conventional boiling method as previously described [57]. All extracted DNA was stored at −20 °C until used for polymerase chain reaction (PCR) amplification. *S. aureus* was confirmed by amplifying a housekeeping gene *nuc*. *S. aureus* was identified as MRSA by amplifying *mecA* and *mecC* genes. PCR was performed in a 25 μL reaction tube with 1 μL template DNA, 2.5 μL 10×*rTaq* Buffer (Mg^2+^Plus) (TaKaRa company, Dalian, China), 2 μL dNTP Mixture (2.5 mM) (TaKaRa company, Dalian, China), 0.125 μL TaKaRa Taq (5 U/μL) (TaKaRa company, Dalian, China), 0.5 μL of each forward and reverse primers, and 18.375 μL ddH_2_O. The following PCR conditions were used; 5 min at 94 °C; 30 cycles of 30 s at 94 °C, 1 min at annealing temperature, and 1 min at 72 °C, and final extension at 72 °C for 10 min. Each PCR was conducted in triplicate using a thermocycler (BioRad, Hercules, CA, USA). PCR products were visualized on 1% agarose gel, stained with ethidium bromide, using electrophoresis for 10 min at 175 volts with 0.5×TBE, and visualized under UV light using the Bio ChemiDoc imaging system (BioRad, California, USA). The *nuc*, *mecA,* and *mecC* genes primers and their positive controls are shown in Appendix A. DNase free water was used as negative controls.

### 4.5. Antimicrobial Susceptibility Testing

All *S. aureus* isolates were investigated for their minimum inhibitory concentrations (MICs) of β-lactams (AMO, OXA, PEN, and FOX), aminoglycosides (GEN), phenicols (FFC), macrolides (ERY), lincosamides (CLI), oxazolidinones (LZD and TZD), pleuromutilins (TIA), fluoroquinolones (CIP) and DOX, TIG, RIF, SXT, VAN, and FOS by broth microdilution following the recommendations given by the Clinical and Laboratory Standards Institute (CLSI) M100-S29 and VET01-A4/VET01-S2 [58,59]. The MIC breakpoints of each antibiotic against *S. aureus* were determined following the recommendations given in the current CLSI guidance [58,59]. Multidrug resistance (MDR) was determined when an isolate was resistant to three or more antibiotics. *S. aureus* ATCC 29213 was used as a quality-control organism.

### 4.6. Molecular Detection of Virulence and Antimicrobial Resistance Genes

All *S. aureus* isolates were further screened by PCR amplification for the presence of virulence and antimicrobial resistance genes: leukocidin genes (*lukS/F-PV*, *lukD* and *lukE*), hemolysin genes (*hla*, *hlb* and *hld*), the toxic shock syndrome toxin gene (*tst*), staphylococcal enterotoxin genes (*sea*, *seb*, *sec*, *sed*, *see*, *seg*, *seh*, *sei*, *sej*, *sek*, *sel*, *sem*, *sen* and *seo*), exfoliative toxin genes (*eta* and *etb*), *ica* operon genes (*icaA*, *icaD*, *icaB*, *icaC* and *icaR*), adhesion genes (*clfA*, *clfB*, *cidA*, *fib*, *fnbA*, *fnbB*, *cna*, *ebps*, *eno*, *sdrC*, *sdrD* and *sdrE*), oxazolidinone resistance genes (*cfr* and *optrA*), vancomycin resistance genes (*vanA* and *vanB*), erythromycin resistance genes (*ermA*, *ermB* and *ermC*), aminoglycoside resistance gene (*aac(6’)-aph(2’’)*), and tetracycline resistance gene (*tetK*). The PCR reaction system is consistent with the reaction system used for molecular identification of *S. aureus*. Each PCR was conducted in triplicate using a thermocycler using a thermocycler (BioRad, California, USA). PCR products were visualized on 1% agarose gel, stained with ethidium bromide, using electrophoresis for 10 min at 175 volts with 0.5×TBE and visualized under UV light using the Bio ChemiDoc imaging system (BioRad, California, USA). All virulence and antimicrobial resistance genes primers and their positive controls are listed in Appendix A. DNase free water was used as negative controls.

### 4.7. Molecular Epidemiology Analysis

The clonality of the *S. aureus* isolates was determined using the *Sma*I PFGE assay as described previously [60]. Comparison of PFGE patterns was performed using BioNumerics software (Applied Maths, Sint-Martens-Latem, Belgium). Dendrograms were generated using Dice similarity coefficient and a similarity cutoff of 100% was used to identify a PFGE cluster. Further identifications of clonality were performed by MLST and *spa* typing as described previously [61,62]. The sequences of seven housekeeping genes (*arcC*, *aroE*, *glpF*, *gmk*, *pta*, *tpi*, and *yqiL*) were submitted to the *S. aureus* MLST database (http://www.saureus.mlst.net/). The sequence of the polymorphic X region of the *spa* gene was submitted to the *S. aureus spa* type database (http://www.spaserver.ridom.de). *Salmonella enterica* serotype Braenderup H9812 DNA was used as a molecular size marker [60].

### 4.8. Identification of Biofilm production Ability of S. aureus

Biofilm production ability was identified using a microtiter plate assay as described previously [63]. *S. aureus* isolates were grown overnight at 37 °C in 2 mL brain heart infusion (BHI) broth plus 0.25% glucose. Cultures were then diluted 1:200 and incubated at 37 °C for 24 h in stationary flat-bottom 96 well polystyrene microtiter plates (Corning Incorporated, New York, NY, USA). Free cells were removed by washing the biofilm three times with sterile 1×PBS, dried in an inverted position, and fixed with 150 μL of ≥99.5% methanol. The methanol was removed and the biofilm were stained with 0.1% crystal violet for 20 min at room temperature. The 96-well plates were rinsed with slow running water to remove excess crystal violet until the water became clear. Crystal violet was dissolved using ethanol/acetone (80:20) for 20 min and the absorbance was measured at 595 nm. The strong biofilm forming strain MRSA ATCC 43300 was used as a positive control. The BHI broth plus 0.25% glucose was used as a negative control. Biofilm production ability was counted as follows: nonbiofilm forming (0 < A595 ≤ 1), weak (1 < A595 ≤ 2), moderate (2 < A595 ≤ 3), strong (A595 > 3). All assays were performed with eight parallels and the results were processed with GraphPad Prism 8.0 software.

### 4.9. Statistical Analysis

Statistical analyses were performed using software SPSS 22.0. Data were analyzed using the chi-square or Fisher’s exact tests. All statistical tests were two-tailed, and *p* < 0.05 or *p* < 0.01 (Fisher’s exact tests among three groups) was deemed statistically significant.

## 5. Conclusions

In summary, our results revealed that *S. aureus* was not only present in veterinarians and pets but also widely distributed in the environment of veterinary hospitals. ST188-t189 and ST398-t571 were the most prevalent in MSSA isolates, while ST59-t437 was the most predominant type of MRSA isolates. Importantly, the emergence of *lukS/F-PV* and *tst* genes in *S. aureus* from veterinary hospitals should raise our vigilance. In addition, the relatively high antibiotic resistance rates of *S. aureus* isolates to PEN and ERY deserved our attention. The biofilm production capacity of *S. aureus* is strongly correlated with ST types and MRSA with strong biofilm production ability often showed greater drug resistance. Some adhesion genes such as *clfA*, *clfB*, *fnbA, and sdrC* may pose an influence on biofilm production ability. Further studies are needed to elucidate the roles of these genes in the biofilm formation of *S. aureus*.

## Figures and Tables

**Figure 1 pathogens-09-00264-f001:**
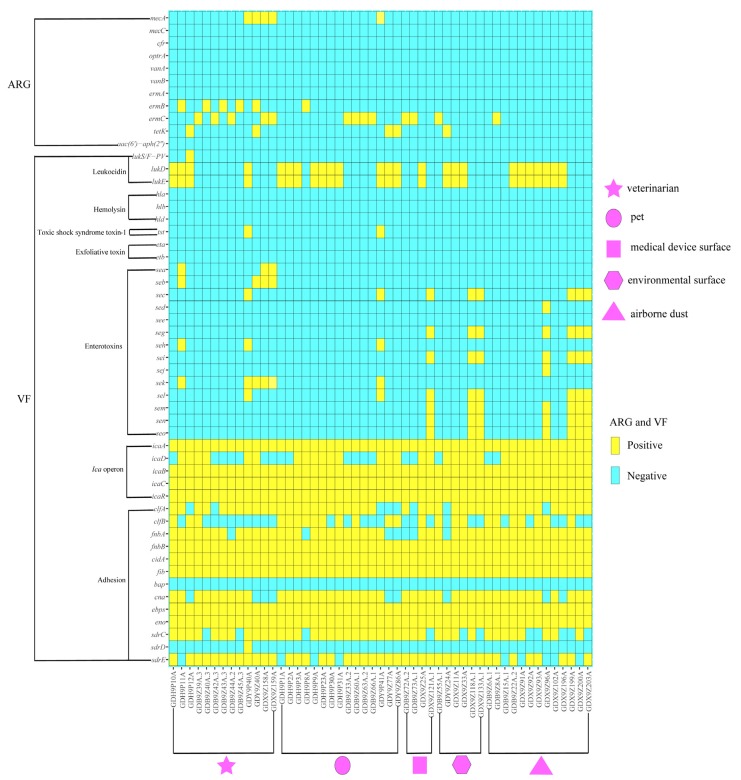
Antibiotic resistance genes and virulence genes profiles in *S. aureus* of different sources from various veterinary hospitals in Guangzhou, China. ARG and VF stand for antimicrobial resistance gene and virulence gene, respectively.

**Figure 2 pathogens-09-00264-f002:**
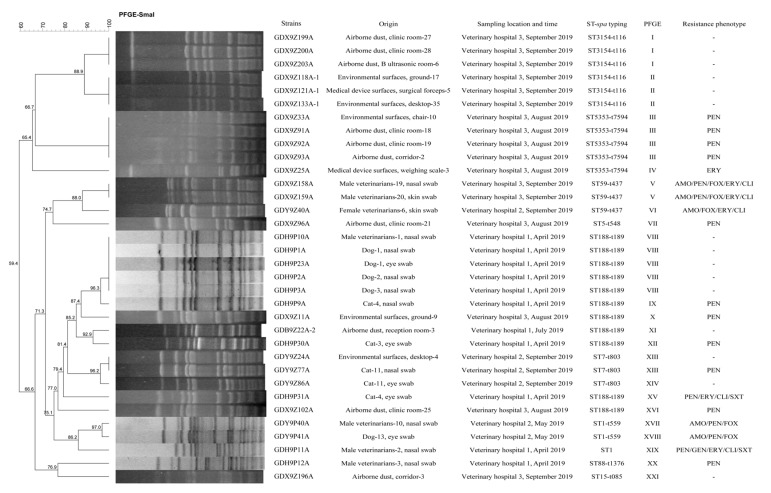
PFGE fingerprint patterns of *S. aureus* isolates together with ST and *spa* types. A similarity cutoff of 100% was used to identify a PFGE cluster. AMO, amoxicillin; PEN, penicillin; FOX, cefoxitin; ERY, erythromycin; CLI, clindamycin; SXT, trimethoprim-sulfamethoxazole; CIP, ciprofloxacin. Resistance phenotype: “–” indicates that *S. aureus* isolates are sensitive to all tested antibiotics.

**Figure 3 pathogens-09-00264-f003:**
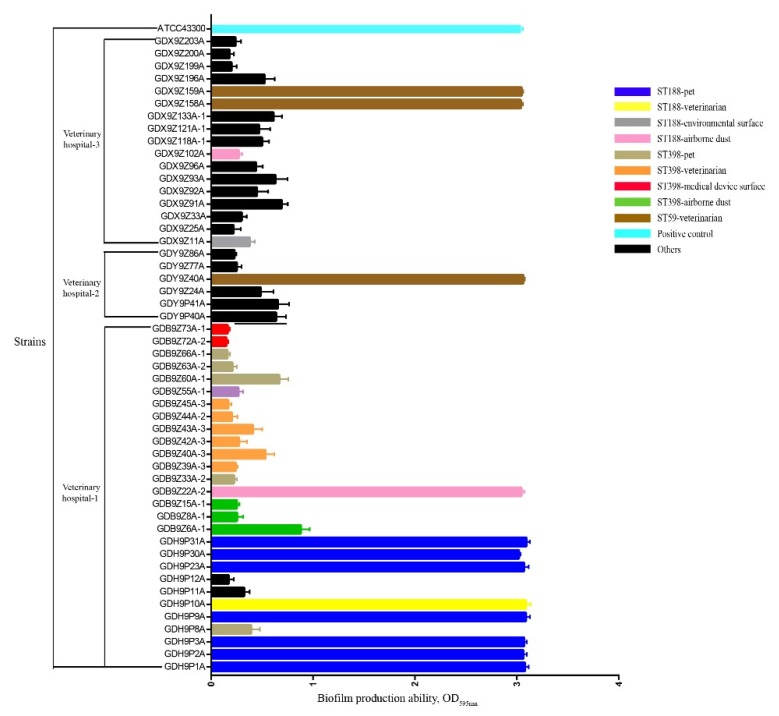
Biofilm production ability of *S. aureus* strains with different STs and sources determined by crystal violet straining assay after being cultured in 96-well plates at 37 °C for 24 h without shaking. The total biofilm formation of *S. aureus* strains including a positive control MRSA ATCC 43300 was measured at OD_595nm_. The data are the averages for eight replicates and the error bars indicate standard deviations (SDs).

**Table 1 pathogens-09-00264-t001:** Prevalence of *Staphylococcus aureus* isolated from veterinary hospitals in Guangzhou, China.

Sampling Location	Veterinary Hospital 1 ^a^	Veterinary Hospital 2 ^b^	Veterinary Hospital 3 ^c^	Total
**Veterinarians**	**9/11**	**2/13**	**2/17**	**13/41**
**Male veterinarians**	**7/7**	**1/7**	**2/7**	**10/21**
Skin swabs	2/2	0/2	1/2	3/10
Nasal swabs	5/5	1/4	1/2	7/11
**Female veterinarians**	**2/4**	**1/6**	**0/10**	**3/20**
Skin swabs	2/2	1/4	0/8	3/14
Nasal swabs	0/2	0/2	0/2	0/6
**Pets**	**11/32**	**3/38**	**0/12**	**14/82**
**Dogs**	**8/24**	**1/14**	**0/6**	**9/44**
Eye swab	4/12	1/7	0/3	5/22
Nasal swab	4/12	0/7	0/3	4/22
**Cats**	**3/8**	**2/24**	**0/6**	**5/38**
Eye swab	2/4	1/12	0/3	3/19
Nasal swab	2/4	1/12	0/3	2/19
**Medical device surfaces**	**2/3**	**0/11**	**2/23**	**4/37**
Infusion pump	-	0/7	0/13	0/20
Weighing scale	-	0/2	1/4	1/6
Surgical forceps	1/2	0/1	1/3	2/6
Surgical scissors	1/1	0/1	0/3	1/5
**Environmental surfaces**	**1/18**	**1/38**	**4/82**	**6/138**
Desktop	0/2	1/13	1/25	2/40
Chair	-	0/8	1/19	1/27
Cage interior	1/14	0/4	0/5	1/23
Ground	0/2	0/4	2/13	2/19
Door handle	-	0/7	0/11	0/18
Switch	-	-	0/7	0/7
Drawer	-	0/2	0/2	0/4
**Airborne dust**	**4/24**	**0/13**	**9/31**	**13/68**
Clinic room	2/12	0/5	6/13	8/30
Inpatient department	1/8	-	0/5	1/13
Hall	-	0/4	0/6	0/10
B ultrasonic room	0/1	0/2	1/4	1/7
Reception room	1/3	0/1	0/1	1/5
Corridor	-	0/1	2/2	2/3
**Total**	**28/88**	**6/113**	**17/165**	**51/366**

^a^ Samples were collected from April 24–26 and July 6–8 in 2019 at the veterinary hospital in Haizhu District, Guangzhou. ^b^ Samples were collected from May 22–24 and September 22–24 in 2019 at the veterinary hospital in Yuexiu District, Guangzhou. ^c^ Samples were collected from August 27–28 and September 21–23 in 2019 at the veterinary hospital in Tianhe District, Guangzhou. The bold numbers represent the isolation rate of *Staphylococcus aureus* from the corresponding samples.

**Table 2 pathogens-09-00264-t002:** Characteristics of *S. aureus* ST398 isolates.

Strains	Origin	Sampling Location and Time	ST-*spa* Typing	Resistance Phenotype
GDH9P8A	Cat-3, nasal swab	Veterinary hospital 1, April 2019	ST398-t571	PEN/ERY/SXT
GDB9Z6A-1	Airborne dust, clinic room-4	Veterinary hospital 1, July 2019	ST398-t034	-
GDB9Z8A-1	Airborne dust, clinic room-6	Veterinary hospital 1, July 2019	ST398-t011	PEN/ERY
GDB9Z15A-1	Airborne dust, inpatient department-4	Veterinary hospital 1, July 2019	ST398-t034	-
GDB9Z33A-2	Dog-7, nasal swab	Veterinary hospital 1, July 2019	ST398-t571	PEN/ERY
GDB9Z39A-3	Male veterinarians-4, nasal swab	Veterinary hospital 1, July 2019	ST398-t571	PEN/ERY
GDB9Z40A-3	Male veterinarians-5, nasal swab	Veterinary hospital 1, July 2019	ST398-t034	PEN/FFC/ERY/CIP
GDB9Z42A-3	Male veterinarians-6, skin swab	Veterinary hospital 1, July 2019	ST398-t571	PEN/ERY
GDB9Z43A-3	Male veterinarians-7, skin swab	Veterinary hospital 1, July 2019	ST398-t034	PEN/FFC/ERY/CIP
GDB9Z44A-2	Female veterinarians-3, skin swab	Veterinary hospital 1, July 2019	ST398-t571	PEN/ERY
GDB9Z45A-3	Female veterinarians-4, skin swab	Veterinary hospital 1, July 2019	ST398-t034	PEN/FFC/ERY/CIP
GDB9Z55A-1	Environmental surfaces, cage interior-10	Veterinary hospital 1, July 2019	ST398-t571	PEN/ERY
GDB9Z60A-1	Dog-6, eye swab	Veterinary hospital 1, July 2019	ST398-t571	PEN/ERY
GDB9Z63A-2	Dog-9, eye swab	Veterinary hospital 1, July 2019	ST398-t571	PEN/ERY
GDB9Z66A-1	Dog-12, eye swab	Veterinary hospital 1, July 2019	ST398-t571	PEN/ERY
GDB9Z72A-2	Medical device surfaces, surgical forceps-2	Veterinary hospital 1, July 2019	ST398-t571	PEN/ERY
GDB9Z73A-1	Medical device surfaces, surgical scissors-1	Veterinary hospital 1, July 2019	ST398-t571	PEN/ERY

Resistance phenotype: “–” indicates that S. aureus isolates are sensitive to all tested antibiotics. PEN, penicillin; FFC, florfenicol, ERY, erythromycin; CIP, ciprofloxacin; SXT, trimethoprim-sulfamethoxazole.

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
