# Peer review of "Biofilm Production Ability, Virulence and Antimicrobial Resistance Genes in Staphylococcus aureus from Various Veterinary Hospitals"

_pathogens, 2020, doi:10.3390/pathogens9040264_

Round 1

Reviewer 1 Report

This version of the paper by Chen et al. is markedly improved.  The authors have addressed all of the concerns that were laid out in the previous draft.  My comments are primarily wording issues as described below.

1.  Line 66  Change to:  India and China being the most dominate consumers.

2.  Line 70 Change rising to rise.

3.  Line 115 Change to handles.

4.  Line 116 Change to halls with airborne dust.

5.  Line 117 Change to different than the female.

6.  Line 143 Change and to or tedizolid.

7.  Line 267  Change to isolates, including MRSA in this study,

8.  A space should be between a number and its unit of measure (e.g. 10 min or 2 mL).

Reviewer 2 Report

The authors now present a manuscript which better reflects the hard work that they have put into this study and is greatly improved from earlier versions.

I am happy that my concerns have been addressed and look forward to the follow up study that the authors have suggested. I did notice a couple of minor things that still need some attention.

Conclusion need to be short and clear 

Please recheck the References order.

Author Response

This manuscript is a resubmission of an earlier submission. The following is a list of the peer review reports and author responses from that submission.

Round 1

Reviewer 1 Report

In general the manuscript "Biofilm Production Ability, Virulence and Antimicrobial Resistance Genes in Staphylococcus aureus from Various Veterinary Hospitals"is good. The topics is really hot. 

I recommend a few modification.

Double check the English.

The discussion section is too long and written more like a review rather than explaining the observations. I would recommend cutting it to make an impact on the reader.

Conclusion need to be short and clear.

Please recheck the References order.

1. in the introduction part " Alarmingly, numerous cases have been reported that humans may be infected with pet-associated S. aureus and pets such as dogs and cats can serve as the reservoir of S. aureus [7-10]" it's not clear rephrase.

2. Paragraph between row 233-246 require to be rephrased.  

3. the Conclusion part can be structured short with 3 short recommendation for clinicians and veterinarians.

Reviewer 2 Report

The paper by Chen et al. describes the biofilm producing ability, virulence factor gene and antibiotic resistance gene profiles of Staphylococcus aureus from three Chinese veterinary hospitals. Much of the data presented will have relevance in terms of discerning the prevalence of S. aureus in these environments.  With that said, several major points need to be addressed to to make the manuscript publishable.

Major criticisms

1.  You cannot make a statement that virulence factor genes were not detected for the majority of the S. aureus strains.  You must screen a number of additional virulence factor genes including the remaining staphylococcal enterotoxin genes, other toxin genes, and particularly adherence factor genes.

2.  A number of details are lacking in your materials and methods.  You need to stipulate that the nuc gene was used as a housekeeping gene.  How many PCR replicates were done?  What positive and negative control strains did you use for all of your genes?

3.  Figure 3 should have the Y and X axis changed.  The present X axis cannot be read and there is no label for what the units mean.  The Y axis also needs a label.  The Figure 3 legend is incomplete.  Remember, a figure is a stand alone entity.  Describe in words what the columns mean.  The data represents the mean + standard deviation or standard error of the mean.  How many replicates were done?

Minor criticisms:

1.  You cannot state that the S. aureus carriage in veterinarians is high (31.7%) when you pointed out that 30% of the population is persistent and 60% are intermittent carriers.

2.  Table 1 should be easier for the reader to interpret.  Bold the main headings (e.g. Veterinarians) and indent the subareas (e.g. skin swabs, nasal swabs).

3.  Spell out all of your acronyms the first time time you use them (e.g. MLST line 21).  The acronyms for the antibiotics need to be explained in the text of the paper, not just in the Table legend.